# The Role of Physical Examination in Assessing Hip Migration in Children with Cerebral Palsy

**DOI:** 10.3390/jcm14165813

**Published:** 2025-08-17

**Authors:** Merel Charlotte Rosalie Roelen, Renée Anne van Stralen, Jim Bono Aalbers, Denise Eygendaal, Max Reijman, Jaap Johannes Tolk

**Affiliations:** Department of Orthopedic Surgery and Sports Medicine, Erasmus MC/Sophia Children’s Hospital, Dr Molewaterplein 40, 3015 GB Rotterdam, The Netherlands; m.roelen@erasmusmc.nl (M.C.R.R.); r.vanstralen-bosboom@erasmusmc.nl (R.A.v.S.); j.aalbers@erasmusmc.nl (J.B.A.); d.eygendaal@erasmusmc.nl (D.E.); m.reijman@erasmusmc.nl (M.R.)

**Keywords:** cerebral palsy, hip migration, migration percentage, abduction in flexion, range of motion

## Abstract

**Background:** Hip migration is a common comorbidity in children with cerebral palsy (CP) and can progress to complete hip dislocation, resulting in a reduced quality of life. Structured surveillance programs designed to prevent complete hip dislocation have demonstrated success regarding the early identification of hip migration. The objectives of this study are to determine whether there is a correlation between hip abduction in flexion (AIF) and migration percentage (MP) and to determine if there are clear cutoff values for hip abduction in flexion that are associated with progressive hip migration. **Methods:** This retrospective study evaluated children at our neuromuscular clinic between 2018 and 2022. We included children diagnosed with spastic CP for whom hip radiographs and concurrent physical examinations were available. The outcomes were assessed using AIF as a measure of range of motion and migration percentage according to Reimers. **Results:** In total, 83 patients were included, with a mean MP of 30.7% and a median AIF of 40 degrees. Mixed-effects modeling revealed a significant correlation between MP and AIF (β = −0.51, *p* < 0.001). Using generalized linear mixed-effects models and ROC analysis, we established a cutoff value of 40 degrees for AIF in predicting MP above 30%, with a sensitivity of 94.5% and a specificity of 80%. **Conclusions:** A negative correlation between AIF and MP was found, indicating that as AIF decreases, MP increases. Furthermore, a distinct cutoff value of 40° for AIF in progressive hip migration was found, which can guide timely referrals and early imaging.

## 1. Introduction

Hip migration is a common comorbidity in children with cerebral palsy (CP) and can progress to complete hip dislocation, resulting in pain, functional impairment, and a reduced quality of life [1]. The risk of hip migration is closely tied to the severity of a child’s neurological impairment, with higher risks observed in those exhibiting greater gross motor dysfunction, as classified by the Gross Motor Function Classification System (GMFCS) [2,3,4,5,6,7]. Without intervention, the overall incidence of hip (sub-)luxation in all cerebral palsy patients ranges from 15 to 30%. This rate increases to up to 90% in patients classified as GMFCS level V [5,8,9,10,11].

Structured surveillance programs designed to prevent hip dislocation in CP patients have demonstrated success regarding the early identification of hip migration [9,12]. These programs facilitate preventive treatments, which have been shown to reduce the rate of hip dislocation to below 0.5% in patients who are medically eligible for surgical intervention [12,13,14,15].

The cornerstone of these surveillance protocols is a regular schedule of physical examinations and hip radiographs. The assessment of hip range of motion, particularly hip abduction in flexion (AIF), is of significance because the prevailing consensus is that hip migration is (partly) attributed to shortened and spastic hip adductor muscles [16,17,18,19]. However, the role of physical examination in decision making in these protocols is limited [20,21]. The necessity for frequent pelvic x-rays is supported by the fact that hip range of motion alone is not an accurate indicator of progressive hip migration [20,21]. Nevertheless, the literature regarding the role of physical examination is limited and contradictory [4,17,22].

Hagglund et al. studied 208 children using the CPUP (Cerebral Pares UppföljningsProgram) registry [4]. They found that measurements of hip range of motion (ROM), specifically hip abduction, at the onset of radiographic hip displacement did not predict significant hip migration. Therefore, they concluded that ROM assessments are inadequate for replacing radiographic screening in detecting hip displacement over time [4].

Metaxiotis et al. analyzed 23 patients with cerebral palsy and unilateral hip migration, comparing physical examinations and pelvic x-rays. They noted a significant difference in hip abduction between migrated and non-migrated sides, but no clear cutoff values could be established [22]. Chang et al. analyzed 31 non-ambulatory patients to see whether hip abduction range correlates with hip displacement. They found that a higher migration percentage was associated with lower hip abduction rates [17].

In the Netherlands, comparable surveillance protocols have been implemented; however, the responsibility for monitoring is primarily delegated to rehabilitation physicians and community physical therapists. Patients are only referred to a pediatric orthopedic surgeon if the migration percentage exceeds 30%. In such cases, the orthopedic surgeon will then take over hip surveillance and determine whether surgical management is indicated. At that point, the primary care team, including the community physical therapist and rehabilitation consultant, will no longer follow up on pelvic radiographs. This is despite them being the team closest to the patient and seeing the patients and their parents most frequently.

Establishing clear cutoff values for hip abduction in flexion using physical examination has the potential to significantly aid general practitioners, community physical therapists, and rehabilitation consultants. These thresholds could allow for clinicians to promptly identify patients at risk for higher hip migration and serve as a adjunctive tool to radiological surveillance to ensure timely referrals for confirmatory imaging. The objective is not to eliminate radiographic imaging from surveillance programs, but rather to explore the potential for physical examination to serve as a complementary component within existing surveillance protocols.

The primary aim of this study was to determine whether there is a correlation between hip abduction in flexion and migration percentage (MP). Our null hypothesis is that there is no correlation. Our secondary aims were to establish clear cutoff values for hip abduction in flexion associated with progressive hip migration and to determine if the velocity of change in hip abduction in flexion is correlated with changes in hip migration.

## 2. Materials and Methods

This retrospective study reviews the records of all children who attended the neuromuscular clinic at Erasmus MC/Sophia Children’s Hospital between 1 January 2018 and 31 December 2022. This study received institutional board approval from Erasmus MC (MEC 2022-0620).

### 2.1. Patient Selection

We reviewed the medical records of all 208 children. Children diagnosed with predominantly spastic cerebral palsy with available hip radiographs and concurrent physical examinations at every visit were included. Children with a history of hip surgery or age above 16 years were excluded, resulting in a sample of 92 children. A complete case analysis was conducted, and, due to missing data for the abduction in flexion variable, a final sample of 83 children was included (Figure 1). For the longitudinal analysis, we were able to include 32 patients who had concurrent radiographs and physical examinations. The remaining 51 patients were either lost to follow-up or received surgery.

### 2.2. Data Extraction

One author (J.A.) conducted a comprehensive medical record review, collecting data on patient age at time of the first visit, sex, GMFCS level, use of tone management (oral baclofen/trihexyfenidyl, intrathecal baclofen, previous selective dorsal rhizotomy), and ROM of the hip (abduction in flexion in degrees) measured by an orthopedic surgeon at all visits.

### 2.3. Measurements

Abduction in flexion measurements were performed by an experienced orthopedic surgeon (R.A.). Abduction in flexion is measured with the patient supine and the hips in 90 degrees of flexion. Both hips are abducted simultaneously (Figure 2). For abduction in flexion, it is known that using a goniometer has excellent intra-rater reliability; however, the inter-rater reliability is more diverse, with some articles reporting excellent reliability, while others report low reliability [23,24].

Radiographic measurements of antero-posterior (AP) pelvis radiographs were performed by J.B. and M.C.R. using Vue PACS (2022 Koninklijke Philips NV, Best, Amsterdam, The Netherlands). The following parameters were assessed: migration percentage (MP), head–shaft angle (HSA), and acetabular index (AI). The migration percentage represents the percentage of the femoral head that is covered by the acetabulum, and this measurement is depicted in Figure 3. This parameter is used to determine the extent of femoral head migration out of the acetabular socket, thereby indicating hip stability [25]. A migration percentage of 30% or more is considered “at risk” for subluxation, while 50% or more is at risk for dislocation [5,13,26,27]. According to the Dutch Guideline, a hip migration percentage of 30% is considered a “hip at risk” and warrants referral to a specialized pediatric orthopedic service [13].

The head-shaft angle, measured in degrees according to Southwick’s method, describes the orientation of the femoral head relative to the femoral shaft. Higher values of this angle are correlated with increased hip displacement [28,29]. The acetabular index, measured in degrees, reflects the inclination of the acetabulum and is important in assessing hip joint development and secondary acetabular dysplasia [30]. Migration percentage and head-shaft angle demonstrate excellent inter-rater and intra-rater reliability for monitoring hip migration [31,32,33]. Acetabular index also demonstrates good inter-rater and intra-rater reliability.

### 2.4. Statistical Analysis

All data analyses were performed using R software version 4.3.3 (R development core team). Significance was defined as *p* < 0.05. We conducted a complete case analysis. Normal distribution was assessed by data visualization using histograms and the Shapiro–Wilk test. Continuous outcome parameters are presented based on their data distribution, while dichotomous measures are reported as counts and percentages.

Our primary research question aims to determine the correlation between hip migration and the degree of hip abduction in flexion observed during physical examination. The primary outcome measure was the migration percentage in relation to the physical examination findings, specifically abduction in flexion. To answer this primary research question, we have fitted a linear mixed-effects model whilst accounting for potential confounding variables such as GMFCS level, multiple measurements per patient, and bilateral hip assessments. This allowed us to estimate the primary question while controlling for the potential confounders. The assumptions of this model were evaluated and confirmed in consultation with a statistician.

We used a generalized linear mixed model to determine a cutoff value for abduction in flexion that predicts an MP above 30%. To evaluate the model’s performance, we utilized a receiver operating characteristic curve, considering the area under the curve, sensitivity, and specificity across various AIF threshold values. Additionally, we used piecewise regression to assess whether the relationship between abduction in flexion and migration percentage changed at the identified cutoff value for abduction in flexion.

Finally, we used a linear mixed-effects model to analyze the velocity of change in hip range of motion and its association with changes in hip migration. This analysis focused on the subgroup of children with multiple clinical visits. The velocity, expressed in percentage points or degrees per year, represents the rate of change in MP and AIF, allowing for a detailed examination of hip migration progression and associated changes in physical examination.

## 3. Results

The mean age of our study population was 7.1 (standard deviation (SD) = 3.6) years at baseline, which corresponds to the first clinical visit. Of these children, 60% were boys. The mean migration percentage was 31%, and the median abduction in flexion was 40 degrees. A total of 83 children were included in our primary analysis and the analysis for an AIF cutoff value. Additionally, 32 children with at least one follow-up visit were included in our velocity analysis, with a median follow-up duration of 1.4 years. All baseline characteristics are presented in Table 1.

A mixed model regression analysis exploring the relationship between MP and AIF revealed a significant negative correlation (β = −0.51; *p* = 5.3 × 10^−10^; 95%; CI: [−6.6%, −3.6%]). This indicates that a 10-degree decrease in AIF corresponds to a 5.1% increase in MP (Figure 4).

Using a generalized linear mixed model, we determined the optimal cutoff value for abduction in flexion, with hip migration percentage above 30% as the outcome. Evaluating the model’s performance across various abduction in flexion threshold values using a receiver operating characteristic curve analysis, a cutoff of 40 degrees AIF demonstrated the optimal balance between high sensitivity (95%) and specificity (80%) (Figure 5 and Table 2).

Subsequently, a piecewise regression model was used to assess the relationship between AIF and MP, considering the identified cutoff of 40 degrees. The analysis revealed a significant change in the relationship between AIF and MP at this threshold. For children with an AIF ≤ 40 degrees, the effect of abduction on hip migration was amplified, with each 10-degree decrease in AIF associated with an 11.5% increase in MP (Figure 6).

Finally, the velocity analysis, examining the rate of change in MP and AIF, demonstrated a significant negative correlation (β: −0.317; *p* = 0.027). This suggests that a decrease in AIF over time is associated with a corresponding increase in MP. This finding is clinically relevant, as it suggests that a sudden reduction in range of motion may serve as an early indicator of progressive hip displacement. Awareness of this relationship could assist primary caregivers and healthcare providers in identifying patients at higher risk, prompting earlier radiographic assessment or referral to a pediatric orthopedic surgeon to evaluate for potential deterioration in hip development.

## 4. Discussion

Our main finding is a significant negative correlation between abduction in flexion and migration percentage. This aligns with observations from Metaxiotis et al., who noted significant differences in hip abduction between migrated and non-migrated sides [22]. Similarly, Chang et al. demonstrated an association between higher MP and lower hip abduction rates [17].

Furthermore, our analysis establishes a clear cutoff value of 40 degrees for AIF, indicating an increased likelihood of MP exceeding 30%. While diminished AIF is often associated with spasticity and contractures, our results suggest that it can also serve as a crucial indicator for potential progressive hip migration. This 40-degree threshold aligns with the findings of Chang et al., further supporting its clinical relevance [17]. However, it is important to acknowledge that Hagglund et al. found hip range of motion at the onset of radiographic hip migration to be an insufficient predictor for migration, highlighting the importance of radiographic screening [4]. Their study differed from ours in several key aspects. First, the physical examinations and X-rays were not necessarily performed on the same day, potentially introducing a time lag of up to a year. Secondly, they used slightly higher threshold for hip migration and examined the relationship across all continuous values of AIF, whereas our study focuses on defining an optimal threshold to identify MP > 30%. These differences may explain the contrasting conclusions regarding the utility of physical examination parameters in predicting hip migration. It is crucial to acknowledge that, while the specificity of 80% for our cutoff value may result in some false-positive cases, in our opinion these are acceptable in the a screening context. Our primary objective is to identify all children who may be developing progressive hip migration, ensuring that no cases are missed. Accepting a small number of false-positive findings allows for the initiation of earlier follow-up care, enabling close monitoring of those who may subsequently require surgical intervention. The measurement of abduction in flexion can be a valuable addition to hip radiographs in the surveillance of progressive hip migration. In settings where children are primarily assessed by physical therapists or rehabilitation physicians who may not have immediate access to pelvic radiographs, AIF measurement can reinforce the decision to refer patients to orthopedic surgeons between follow-up appointments.

Furthermore, our study found that a more rapid decline in abduction in flexion correlates with a quicker increase in migration percentage. Previous research by Wagner and Hagglund in 2022 examined the concept of MP velocity, investigating the relationship between age and hip migration [34]. They observed that MP increased in the years preceding preventive surgery, and the velocity of MP decreased as children aged, concluding that migration exhibits a high velocity before surgery that diminishes over time. However, their analysis focused solely on the yearly rate of change in MP, without examining the corresponding fluctuations in AIF. Our findings suggest that, as AIF declines, MP tends to increase, which can hold important clinical implications for the surveillance and management of hip migration.

The value of hip surveillance and monitoring is widely established, playing a crucial role in preventing symptomatic hip dislocations [9,10,12,13]. While our findings highlight a strong relationship between abduction in flexion and migration percentage, they should not be interpreted as a basis for replacing radiographic screening. Rather, AIF should be considered a valuable component within a broader set of predictors for MP. Integrating physical examinations into these guidelines can be valuable, especially in settings where routine radiographic imaging may be less accessible, such as general practice, rehabilitation centers, and resource-limited countries.

In practice, children with an AIF of ≤40 degrees could be referred more quickly to an orthopedic surgeon for imaging, comprehensive examination, and treatment. Additionally, in clinical scenarios where AIF shows rapid decline in between radiological surveillance, earlier radiographic imaging may be warranted and could be achieved by monitoring AIF. The aim is not to significantly modify the current surveillance programs, but rather to incorporate physical examination as an additional component in monitoring children with spastic cerebral palsy and risk of hip dislocation.

Future research should examine the incorporation of abduction in flexion into predictive models for progressive hip migration, as this may enhance the models’ predictive capabilities. To facilitate more accurate predictive models, larger data sets would be beneficial to enable more robust predictions. Additionally, investigating other potential predictive factors for (progressive) hip migration, such as head-shaft angle and acetabular index, should be performed to improve the accuracy of these models.

### Limitations

This study has several limitations that should be considered. Due to the limited longitudinal data available, with most children having only a single baseline measurement, our ability to draw definitive conclusions about the long-term relationship between AIF and MP was restricted. Additionally, our longitudinal analysis has an important limitation to consider. Because our study excluded children who had previously undergone surgery, the hips included in the longitudinal analysis were primarily those with less severe migration. By excluding children who had previously undergone surgery, our longitudinal analysis focused primarily on hips with less severe migration. This is because children with higher initial migration were more likely to have already undergone surgery, resulting in their data being excluded from this specific analysis. This selection bias may have led to an underestimation of the correlation between AIF decline and MP increase. Similarly, as children are typically referred when their migration percentage exceeds 30%, this might impact the variation in children included in this cohort. Although the tertiary referral center setting of our study may limit the generalizability of our findings to other clinical settings, it is important to note that, in the Netherlands, the majority of cerebral palsy patients with orthopedic issues are referred to specialized centers. As such, the influence of the study setting on the applicability of our results may be less substantial than one might initially assume. Another potential limitation is the possibility of low inter-rater reliability in measuring abduction in flexion, meaning that different clinicians might obtain slightly different measurements for the same patient. This limitation can be mitigated by ensuring that measurements are consistently obtained with the use of a goniometer.

## 5. Conclusions

This study demonstrates a significant negative correlation between abduction in flexion and hip migration percentage. It establishes a distinct threshold of 40 degrees for abduction in flexion, whereas values below this threshold indicate an increased likelihood of migration percentage exceeding 30%. This finding can aid timely referrals and early imaging, in addition to the current surveillance protocols. These findings do not negate the need for routine radiographic screening according to the well-established guidelines. Although these results do not replace radiographic screening, abduction in flexion may serve as a valuable adjunctive measure for monitoring and managing hip migration in this population, particularly for physiotherapists and primary care providers who are familiar with the patients and see them regularly.

## Figures and Tables

**Figure 1 jcm-14-05813-f001:**
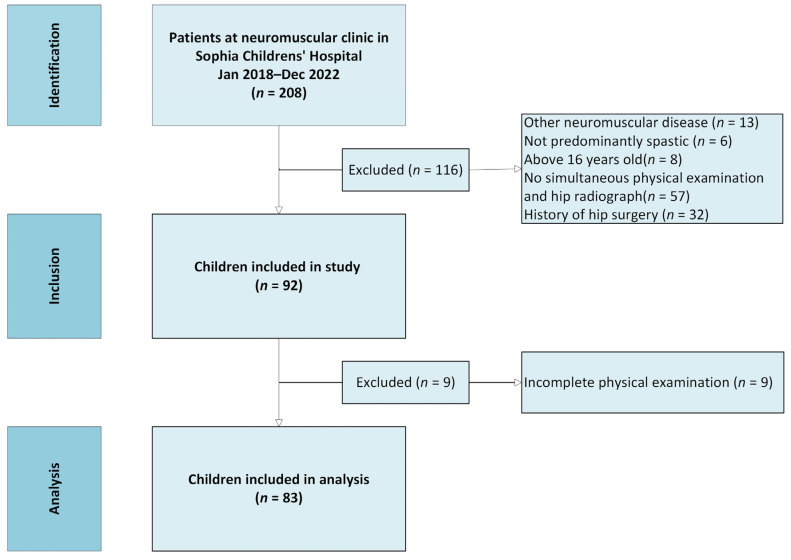
Flow chart of patient selection.

**Figure 2 jcm-14-05813-f002:**
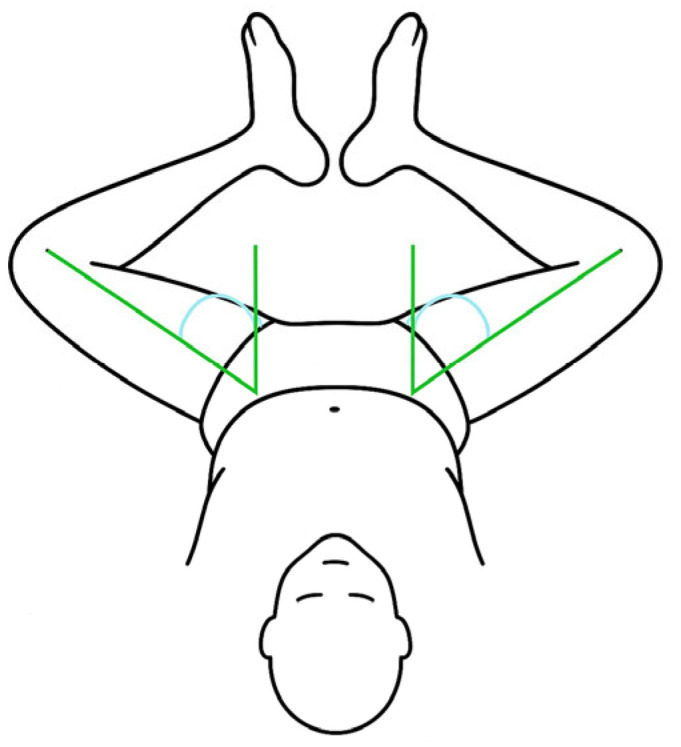
Measurement of hip abduction in flexion. Both hips remain in 90° of flexion. The green and blue lines depict the angle measured.

**Figure 3 jcm-14-05813-f003:**
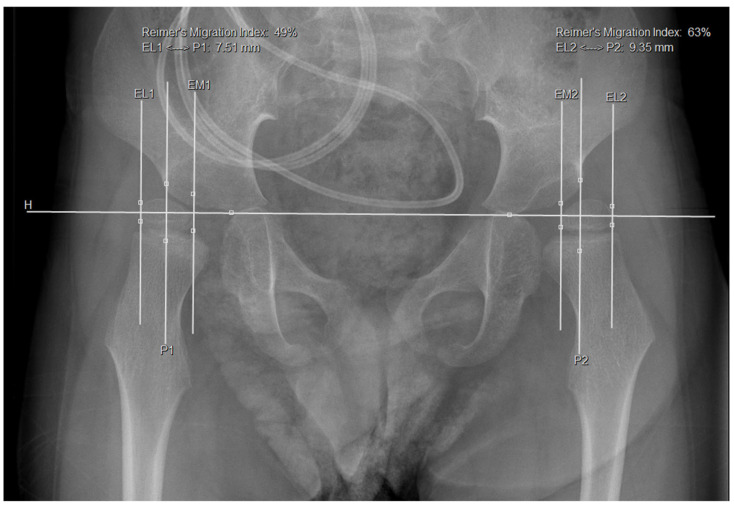
This picture shows an antero-posterior pelvic radiograph with measurements of the migration percentage. P1 is Perkin’s line on the right and p2 is Perkin’s line on the left. EL1 is the most lateral aspect of the epiphysis on the right and EM1 is the most medial aspect of the epiphysis of the femoral head. The MP is then calculated by dividing the distance between EL1 and P1 by the total width of the femoral epiphysis (EL1-EM1). EL2 and EM2 are the corresponding measurements on the left hip. In this specific radiograph, the MP is 49% on the right and 63% on the left hip.

**Figure 4 jcm-14-05813-f004:**
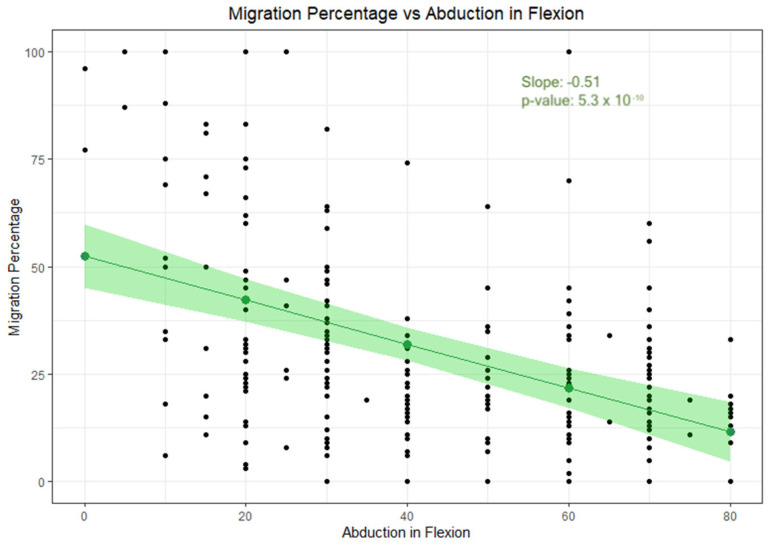
Relationship between migration percentage and abduction in flexion. The confidence interval is marked by the colored area.

**Figure 5 jcm-14-05813-f005:**
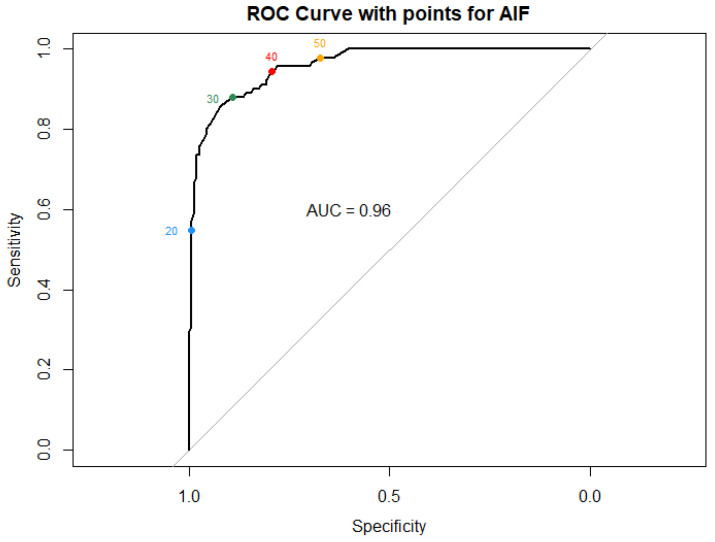
Receiver operating characteristic curve comparing different abduction in flexion cutoff values. Colored numbers represent each value of abduction assessed as a threshold.

**Figure 6 jcm-14-05813-f006:**
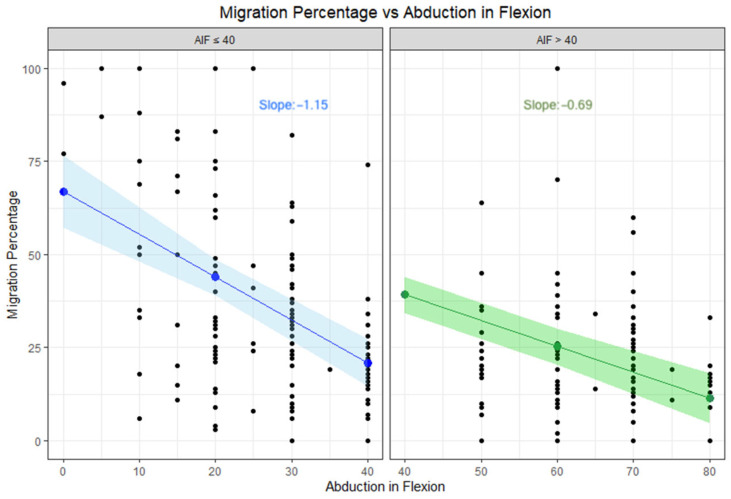
Migration percentage vs. abduction in flexion with a cutoff value of 40 degrees. The confidence interval is marked by the colored area.

**Table 1 jcm-14-05813-t001:** Characteristics of study population.

Included Patients, *n*	83
Included hips, *n*	163
Age, years (SD)	7.1 (3.6)
Boys, *n* (%)	50 (60)
GMFCS ^1^ level, *n* (%)	
I.	13 (16)
II.	9 (11)
III.	17 (20)
IV.	31 (37)
V.	13 (16)
Tone management, *n* (%)	
Yes	23 (28)
Median follow-up, years (IQR ^2^)	1.4 (0.97; 2.3)
Radiological and physical examination parameters	
MP ^3^, percentage (SD)	31 (24)
HSA ^4^, degrees (SD)	163 (11)
AI ^5^, degrees (SD)	20 (7)
AIF ^6^, degrees, median (IQR)	40 (27.5; 60)

^1^ GMFCS: Gross Motor Function Classification System. ^2^ IQR: interquartile range. ^3^ MP: migration percentage. ^4^ HSA: head shaft angle. ^5^ AI: acetabular index. ^6^ AIF: abduction in flexion.

**Table 2 jcm-14-05813-t002:** Specificity and sensitivity for different thresholds of abduction in flexion.

AIF Value	Sensitivity	Specificity
20	0.55	0.99
30	0.88	0.89
40	0.95	0.80
50	0.98	0.67

## Data Availability

The data presented in this study are available on reasonable request from the corresponding author. The data are not publicly available due to security and confidentiality concerns.

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
