# Peer review of "The Role of Physical Examination in Assessing Hip Migration in Children with Cerebral Palsy"

_jcm, 2025, doi:10.3390/jcm14165813_

Round 1

Reviewer 1 Report

Comments and Suggestions for Authors

I commend the authors for their thoughtful and well-structured manuscript titled "The Role Of Physical Examination In Assessing Hip Migration In Children With Cerebral Palsy." This original research addresses an important and timely clinical question—whether a correlation exists between hip abduction in flexion and the migration percentage (MP) on radiographs in children with cerebral palsy (CP). The study is relevant to clinicians involved in neuromuscular orthopedic care and contributes meaningfully to the discussion on non-radiographic screening tools in CP hip surveillance.

The manuscript is generally well-written, clear, and logically organized. The Introduction offers a concise and informative background, framing the need for improved clinical predictors of hip migration. The Materials and Methods section adequately describes the study design, inclusion criteria, and statistical analysis. The Results are clearly presented, supported by two tables and three well-designed figures that improve comprehension. The Discussion is appropriately referenced and contextualizes the findings within current literature. Importantly, the authors transparently acknowledge the study’s limitations.

However, a few minor clarifications and additions would enhance the manuscript’s methodological transparency and clinical utility:

Clinical Measurements:

  • Please clarify how hip abduction in flexion was measured. Was hip flexion standardized to 90 degrees during the assessment?
  • What was the position of the contralateral leg during the measurement? (e.g., extended, flexed, abducted, etc.)
  • If feasible, a clinical photograph or diagram demonstrating the measurement technique would significantly improve reader understanding and reproducibility.

Radiographic Measurements:

  • Consider including a representative anteroposterior pelvic radiograph demonstrating a migration percentage (MP) of approximately 40%, with measurement lines superimposed. This would aid readers in visualizing how MP was determined and serve as a reference point for interpretation.

These suggestions are intended to improve the clarity and applicability of the study, particularly for clinicians who may wish to replicate the physical examination technique or interpret its findings in practice.

In conclusion, this is a valuable and clinically relevant manuscript.

Author Response

Response to the editor,

We would like to express our sincere appreciation to the editor-in-chief and the two reviewers for their careful consideration of our manuscript. We are pleased to have the chance to revise our manuscript and enhance it based on your feedback.

 Reviewer 1

Comments and Suggestions for Authors

I commend the authors for their thoughtful and well-structured manuscript titled "The Role Of Physical Examination In Assessing Hip Migration In Children With Cerebral Palsy." This original research addresses an important and timely clinical question—whether a correlation exists between hip abduction in flexion and the migration percentage (MP) on radiographs in children with cerebral palsy (CP). The study is relevant to clinicians involved in neuromuscular orthopedic care and contributes meaningfully to the discussion on non-radiographic screening tools in CP hip surveillance.

The manuscript is generally well-written, clear, and logically organized. The Introduction offers a concise and informative background, framing the need for improved clinical predictors of hip migration. The Materials and Methods section adequately describes the study design, inclusion criteria, and statistical analysis. The Results are clearly presented, supported by two tables and three well-designed figures that improve comprehension. The Discussion is appropriately referenced and contextualizes the findings within current literature. Importantly, the authors transparently acknowledge the study’s limitations.

Thank you so much for your kind words and time and effort spent on reviewing our paper.

However, a few minor clarifications and additions would enhance the manuscript’s methodological transparency and clinical utility:

Clinical Measurements:

  • Please clarify how hip abduction in flexion was measured. Was hip flexion standardized to 90 degrees during the assessment?

Response:       Yes, Hip flexion was measured with 90 degrees of hip flexion.

Change:            We have adjusted this in the manuscript and added figure 2 to further

clarify this.

  • What was the position of the contralateral leg during the measurement? (e.g., extended, flexed, abducted, etc.)

Response:       Both hips are abducted simultaneously.

Change:            We have added this in the text and hope to have clarified this further by the introduction of figure 2.

  • If feasible, a clinical photograph or diagram demonstrating the measurement technique would significantly improve reader understanding and reproducibility.

Response:       We have added a diagram of the measurement as figure 2.

Change:            See the introduction of figure 2

Radiographic Measurements:

  • Consider including a representative anteroposterior pelvic radiograph demonstrating a migration percentage (MP) of approximately 40%, with measurement lines superimposed. This would aid readers in visualizing how MP was determined and serve as a reference point for interpretation.

Response:       We have added this as Figure 3.

Change:            Introduction of Figure 3.

These suggestions are intended to improve the clarity and applicability of the study, particularly for clinicians who may wish to replicate the physical examination technique or interpret its findings in practice.

In conclusion, this is a valuable and clinically relevant manuscript.

We appreciate the time and expertise invested by the reviewers and editorial team in evaluating our work. We trust that the revisions and clarifications provided have addressed all concerns, and we remain happy to provide any additional information or explanation that may be helpful in further consideration of our manuscript.

Reviewer 2 Report

Comments and Suggestions for Authors

The abstract is structured; clarify the clinical implications of the 94.5% sensitivity and 80% specificity upfront; the keywords should be checked in accordance with MeSH.

The introduction provides a thorough background on hip migration in CP, including GMFCS-related risks and surveillance programs

The methods are well structured in subsection. Justify the choice of 30% MP as the threshold for "at-risk" hips with references. How were confounding variables (e.g., GMFCS levels, tone management) were controlled in the mixed models? Please provide the manufacturer and country of the statistical software used.

In the results section please report interrater reliability metrics for AIF measurements if available. Also, discuss the clinical significance of the velocity analysis (-0.317) in more depth.

In the discussion section please provide more information rehabilitation protocols, in relation to the scientific literature (for e.g. https://doi.org/10.1016/j.promfg.2018.03.122). Limitations are provided at the end. Limitations are provided at the end.

In the conclusions reiterate the importance of combining AIF with MP rather than replacing imaging.

Editing recommendation – the references should be noted with “[ ]” rather than “( )”; number also the subsections.  

Author Response

Response to the editor,

We would like to express our sincere appreciation to the editor-in-chief and the two reviewers for their careful consideration of our manuscript. We are pleased to have the chance to revise our manuscript and enhance it based on your feedback.

Reviewer 2:

Comments and Suggestions for Authors

The abstract is structured; clarify the clinical implications of the 94.5% sensitivity and 80% specificity upfront; the keywords should be checked in accordance with MeSH.

The introduction provides a thorough background on hip migration in CP, including GMFCS-related risks and surveillance programs

The methods are well structured in subsection. Justify the choice of 30% MP as the threshold for "at-risk" hips with references.

Response:       We appreciate the reviewer’s observation. This point is already addressed in lines 128–129 of the manuscript; however, we have now added further clarification in this section to ensure the statement is more explicit.

Change:            We have adjusted these lines to the following:

A migration percentage of 30% or more is considered "at risk" for subluxation, while 50% or more is at risk for dislocation. (5, 13, 26, 27) According to the Dutch Guideline, a hip migration percentage of 30% is considered a “hip at risk” and warrants referral to a specialized pediatric orthopedic service (13)

How were confounding variables (e.g., GMFCS levels, tone management) were controlled in the mixed models?

Response:       The mixed-effects model incorporated potential confounders, including GMFCS level, repeated measurements per patient, and bilaterality, allowing the effect of the primary predictor to be estimated independently of these variables by holding them constant in the analysis. This is described in line 156-160.

Change:            We have added the following line to line 159-160.

This allowed us to estimate the primary question while controlling for the potential confounders.

Please provide the manufacturer and country of the statistical software used.

Response:       R is not manufactured in a single country but developed by an international group.

Change:            We have added the manufacturing group to the text.

In the results section please report interrater reliability metrics for AIF measurements if available.

Response:       Because this study is a retrospective cohort derived from routine clinical practice, interrater reliability for the measurement of abduction in flexion could not be calculated; all assessments were performed by a single consultant.

Change:            -

Also, discuss the clinical significance of the velocity analysis (-0.317) in more depth.

Response:       Thank you for your feedback.

Change:            We have added the following explanation.

This finding is clinically relevant, as it suggests that a sudden reduction in range of motion may serve as an early indicator of progressive hip displacement. Awareness of this relationship could assist primary caregivers and healthcare providers in identifying patients at higher risk, prompting earlier radiographic assessment or referral to a pediatric orthopedic surgeon to evaluate for potential deterioration in hip development.

In the discussion section please provide more information rehabilitation protocols, in relation to the scientific literature (for e.g. https://doi.org/10.1016/j.promfg.2018.03.122).

Response:       Whilst we acknowledge the importance of rehabilitation protocols with regard to function and range of motion in children with CP, we feel that including the cited paper on "mechatronic system for gait rehabilitation" could lead to confusion and divert from the focus of the paper.

Change:            -

Limitations are provided at the end. Limitations are provided at the end.

In the conclusions reiterate the importance of combining AIF with MP rather than replacing imaging.

Response:       Thank you for your feedback, we think this is a very valuable comment.

Change:            We have changed the word “Guide” to “aid”. Next to that we have changed the following sentence to:

These findings do not negate the need for routine radiographic screening according to the well established guidelines. Although these results do not replace radiographic screening, abduction in flexion may serve as a valuable adjunctive measure for monitoring and managing hip migration in this population, particularly for physiotherapists and primary care providers who are familiar with the patients and see them regularly.

Editing recommendation – the references should be noted with “[ ]” rather than “( )”; number also the subsections.  

Response:       We have changed this throughout the text.

Change:            We have changed this throughout the text.

We appreciate the time and expertise invested by the reviewers and editorial team in evaluating our work. We trust that the revisions and clarifications provided have addressed all concerns, and we remain happy to provide any additional information or explanation that may be helpful in further consideration of our manuscript.
